# One-to-Multiple: A Progressive Style Transfer Unsupervised Domain-Adaptive Framework for Kidney Tumor Segmentation

**Kai Hu**
Xiangtan University
kaihu@xtu.edu.cn

**Jinhao Li**
Xiangtan University
jhli@smail.xtu.edu.cn

**Yuan Zhang**[*]
Xiangtan University
yuanz@xtu.edu.cn

**Xiongjun Ye**
Cancer Hospital, Chinese Academy of Medical Sciences
yexiongjun@cicams.ac.cn

**Xieping Gao**[*]
Hunan Normal University
xpgao@hunnu.edu.cn

## Abstract

In multi-sequence Magnetic Resonance Imaging (MRI), the accurate segmentation of the kidney and tumor based on traditional supervised methods typically necessitates detailed annotation for each sequence, which is both time-consuming and labor-intensive. Unsupervised Domain Adaptation (UDA) methods can effectively mitigate inter-domain differences by aligning cross-modal features, thereby reducing the annotation burden. However, most existing UDA methods are limited to one-to-one domain adaptation, which tends to be inefficient and resource-intensive when faced with multi-target domain transfer tasks. To address this challenge, we propose a novel and efficient *One-to-Multiple* **P**rogressive **S**tyle **T**ransfer **U**nsupervised **D**omain-**A**daptive (PSTUDA) framework for kidney and tumor segmentation in multi-sequence MRI. Specifically, we develop a multi-level style dictionary to explicitly store the style information of each target domain at various stages, which alleviates the burden of a single generator in a multi-target transfer task and enables effective decoupling of content and style. Concurrently, we employ multiple cascading style fusion modules that utilize point-wise instance normalization to progressively recombine content and style features, which enhances cross-modal alignment and structural consistency. Experiments conducted on the private MSKT and public KiTS19 datasets demonstrate the superiority of the proposed PSTUDA over comparative methods in multi-sequence kidney and tumor segmentation. The average Dice Similarity Coefficients are increased by at least 1.8% and 3.9%, respectively. Impressively, our PSTUDA not only significantly reduces the floating-point computation by approximately 72% but also reduces the number of model parameters by about 50%, bringing higher efficiency and feasibility to practical clinical applications. [2]

## 1 Introduction

Kidney tumor segmentation is a crucial task in medical image analysis, playing an essential role in the diagnosis, staging, and treatment of kidney cancer [1, 2, 3]. Previous studies have primarily focused on computed tomography (CT) images [4, 5, 6, 7, 8, 9], but in recent years, magnetic resonance imaging (MRI) has emerged as a safer alternative due to its non-radiative nature [10, 11, 12]. Compared to CT,

---

[*]Corresponding authors.
[2]Code is publicly available at: https://github.com/MLMIP/PSTUDA.

38th Conference on Neural Information Processing Systems (NeurIPS 2024).

MRI offers superior contrast for characterizing kidney tissue and tumors, enabling a clearer distinction between normal and pathological tissues. Furthermore, the multi-parameter imaging capability of MRI generates diverse sequences [13, 14, 15] that enhances the comprehensive description of pathological features. These advantages have established MRI as the preferred modality for clinicians in diagnosing kidney disease, particularly kidney tumors, and have facilitated the widespread use of multi-sequence MRI in clinical practice for kidney and tumor segmentation.

In existing studies, traditional supervised segmentation methods typically rely on a large amount of annotated data, and performing fine-grained annotations for a single sequence is already a time-consuming and labor-intensive task, let alone for every sequence. Unsupervised domain adaptation (UDA) methods [16, 17, 18, 19, 20] have been proposed as a promising solution to address these challenges. Current UDA methods mainly focus on one-to-one domain adaptation, involving a single source domain and a target domain. Although these methods perform well in certain scenarios, they lack scalability, meaning that for multi-target domain tasks, training paired generators for each source-target domain pair results in significant computational costs and resource consumption. Furthermore, one-to-one domain adaptation methods can only learn fixed mappings between two domains, thereby failing to capture the potential commonalities and connections among multiple sequences.

As an initial attempt to apply multi-domain adaptation technology to medical imaging, Xu et al. [21] proposed OMUDA for abdominal organ segmentation using multi-sequence MRI. However, the architecture of OMUDA is derived from StarGAN v2 [22], which was originally designed for style transfer in natural images and has not been specifically tailored and optimized for the needs of domain adaptation in medical imaging. Although it shows improvement in resolving domain confusion across multiple domains, this limitation leads to suboptimal generated images with regards to organ structural consistency and detail preservation. Consequently, the task of one-to-multiple domain adaptation in medical imaging remains a significant challenge, necessitating more refined and specialized approaches for further advancement.

In this paper, we propose a *One-to-Multiple* **P**rogressive **S**tyle **T**ransfer **U**nsupervised **D**omain-**A**daptive (PSTUDA) framework, which explicitly stores multi-level style features from different domains in designated multi-level style dictionaries, thus alleviating the burden on the generator and achieving the decoupling of content features from style features. Our PSTUDA employs multiple cascaded style fusion modules to recombine multi-level style features from different domains with content features layer by layer using **P**oint-wise **I**nstance **N**ormalization (PIN), thereby ensuring that the generated images across multiple target domains have high-quality style and structure.

The main contributions of our work are summarised as follows:

- We explore a novel and efficient *One-to-Multiple* **P**rogressive **S**tyle **T**ransfer **U**nsupervised **D**omain-**A**daptive (PSTUDA) framework, which is capable of simultaneously transferring a single annotated source domain to multiple unannotated target domains, significantly reducing the need for tedious domain adaptation work for each target domain.

- We introduce a multi-level style dictionary that stores style information for each domain at different stages of style transfer, alleviating the burden on the generator to perform multiple tasks and effectively decoupling content features from style features.

- We propose a progressive style transfer paradigm and a **P**oint-wise **I**nstance **N**ormalization (PIN) method. The former comprises multiple cascading style fusion modules, each recombining content features with corresponding style features through PIN, thereby achieving better cross-modal alignment and structural consistency.

- We construct a multi-sequence kidney tumor MRI dataset called MSKT to facilitate research on kidney tumor analysis. Quantitative and qualitative results on the MSKT and the public dataset KiTS19 show that our PSTUDA framework outperforms the state-of-the-art methods and significantly improves segmentation performance and training efficiency.

## 2 Related Work

### 2.1 Kidney Tumor Segmentation

Computer-aided diagnostic methods for kidney tumor segmentation play a crucial role in clinical practice [23, 24, 25, 26, 27]. The significant variability in the size, shape, and location of kidneys and

tumors presents a considerable challenge to accurate segmentation. Yu et al. [28] developed Crossbar-Net to capture global and local features of kidney tumors through crossbar patches and focused on difficult-to-segment regions through a cascade training strategy. Myronenko et al. [29] designed a dedicated boundary branch supervised by an edge-aware loss term to enhance the consideration of organ and tumor edge information. These approaches are centered on improving the ability of the model to recognize the complex morphology of kidney tumors. To further advance this effort, Pandey et al. [30] integrated active contouring with 3D-U-Net to achieve precise delineation of kidney tumor shapes, showcasing the potential of merging deep learning with traditional image processing techniques. However, these approaches typically rely on fully supervised learning with extensive pixel-level annotations. To address this limitation, researchers are exploring alternative solutions in the fields of semi-supervised, self-supervised, and unsupervised learning. Ma et al. [31] introduced an Affinity Network that learns from limited data using k-nearest neighbors attention pooling layers. Similarly, Ciga et al. [32] and Faust et al. [33] developed methods that enhance feature learning and guide tumor analysis through self-supervised and unsupervised techniques, respectively. By reducing the reliance on annotated data, these methods not only lower costs but also improve the model's generalization capabilities.

## 2.2   Unsupervised Domain Adaptation

UDA is one of the important methods for addressing domain difference. It aims to transfer a model from an annotated source domain to an unannotated target domain. Existing works have mostly focused on one-to-one domain adaptation [34, 35, 36, 37, 38, 39], yielding impressive outcomes. For example, CycleGAN [34] used cycle consistency constraints to transform unpaired images from one domain to another. CyCADA [40] enforced cycle consistency by combining methods of image space alignment and latent representation space alignment. MUNIT [41] decomposed image representations into content and style codes, enabling multimodal image translation. For medical image domain adaptation, SIFA [35] achieved domain alignment from both image and feature perspectives, enabling the segmentation network to adapt to the unannotated target domain. Thereafter, DEPL [39] further improved the segmentation accuracy by employing multi-source domain extension and selective pseudo-labelling strategies.

However, these one-to-one domain adaptation methods lack scalability when handling multiple domain transfer tasks, as they can only learn the relationships between two different domains at a time. StarGAN [42] performed image-to-image translation across multiple domains using a single model, and then the improved version, StarGAN v2 [22] further enhanced diversity in generated images by introducing style codes specific to each domain. Additionally, Sharma and Hamarneh [43] proposed a multi-modal generative adversarial network that leveraged redundant information from available sequences to synthesize missing MRI pulse sequences in patient scans. Gholami et al. [44] developed an information-theoretic approach that aimed to find shared latent spaces for domain adaptation across multiple target domains. It should be noted that these methods predominantly focus on the diversity of the generated images, while paying less attention to the structural consistency before and after image translation.

## 2.3   Normalization of Image Translation

In UDA tasks, image normalization is a key step that can help models to learn and transfer features effectively [45, 46, 47, 48, 49, 50]. Instance Normalization (IN) [51] normalizes the features of each sample to improve the quality and realism of generated images. Adaptive Instance Normalization (AdaIN) [52] dynamically adjusts the relationship between style and content features of input images to enable a rapid transformation across arbitrary styles. Batch-Instance Normalization (BIN) [53] explicitly normalizes unnecessary style variations in images while preserving useful styles. AdaAttN [54] introduces adaptive attention normalization to optimize the effects of arbitrary neural style transfer. GramLIN [55] continuously measures the proximity of the current stylized output to the target style to achieve progressive stain transfer. In medical image domain adaptation tasks, it is crucial to maintain structural consistency during image translation, in addition to changing image styles. Therefore, we propose a novel normalization method called PIN, which progressively fuses content and style features at each pixel by considering image details and local style differences. The goal is to ensure well-structured images after transfer.

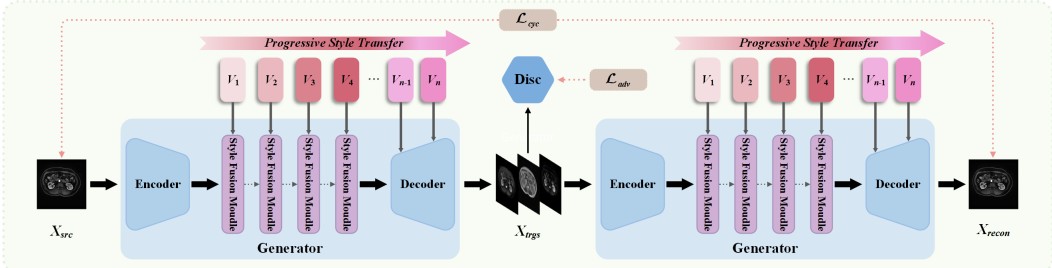

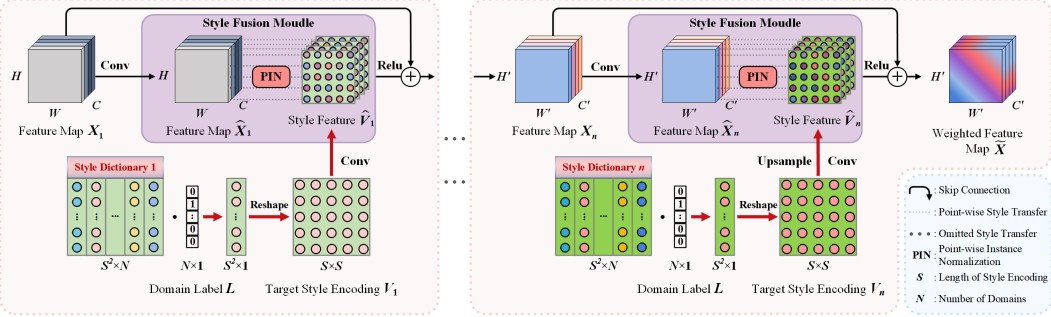

(a) Overview of the proposed One-to-Multiple Progressive Style Transfer Unsupervised Domain-Adaptive framework.

(b) Illustration of the progressive style transfer process.

Figure 1: (a) The overall architecture of the proposed One-to-Multiple Progressive Style Transfer Unsupervised Domain-Adaptive framework, which includes a shared generator and a discriminator. The generator is composed of an encoder, a decoder, and multiple style fusion modules. (b) shows the progressive style transfer process, achieved through cascaded style fusion modules.

## 3  Method

### 3.1  Overview of the proposed One-to-Multiple Framework

Formally, for a one-to-multiple domain adaptation task, we have one source domain $S$ and multiple target domains $T_i$. The source domain $S$ is annotated, denoted as $S = \{(x_j, y_j)\}_{j=1}^{N_s}$, while the target domains are unannotated, denoted as $T_i = \{x_j^{T_i}\}_{j=1}^{N_{T_i}}$. Each domain is assigned a domain label, for example, $L = [0, 1, ..., N]$ representing the source domain 0, target domain 1, *etc.*, up to target domain *N*. Figure 1(a) illustrates our PSTUDA framework, which mainly consists of a shared generator (including an encoder, a decoder, and multiple style fusion modules) and a multi-scale discriminator. $X_{src}$, $X_{trgs}$, and $X_{recon}$ represent the input image from the source domain, the generated pseudo-target domain images corresponding to the input image, and the image reconstructed back to the source domain from the pseudo-target domain images, respectively. Our task involves two stages. The goal of the first stage is to train a single generator $G$ such that given a source domain image $X_{src} \in S$ and any target domain label $l_i$, it can generate a pseudo-target domain image $X_{trg\_i}$ corresponding to the image $X_{src}$, *i.e.*, $G(X_{src}, l_i) \Rightarrow X_{trg\_i}$. The objective of the second stage is to utilize the generated pseudo-target domain images $X_{trgs}$ and source domain annotation $y$ to train a segmentation network to achieve accurate segmentation of the kidney and tumor.

### 3.2  Multi-level Style Dictionary

Our design is inspired by the StarGAN v2 multi-domain image generation framework, which employs a Mapping Network and a Style Encoder to obtain style encodings. The Mapping Network transforms random Gaussian noise into diverse style encodings, while the Style Encoder extracts style encodings from reference images. These techniques are essential for transforming natural image styles and enhancing artistic variety. However, in medical image domain adaptation, they encounter challenges in extracting comprehensive style features. Style encodings from a single image may not capture the full range of styles within a domain, and deriving styles from Gaussian noise, while introducing

diversity, can result in instability. In medical image analysis, such instability could potentially compromise feature recognition and segmentation accuracy.

To address these challenges, we propose to define a set of learnable multi-level style dictionaries for each domain, where the style information at each level closely corresponds to the respective phase of style fusion. This multi-level structure allows for starting with basic style features and exploring more complex layers of style, thereby providing a progressively refined path for the style fusion process. During the generative adversarial process, the continuous updating and learning of the style dictionary can adapt to the ever-changing style demands. The early-stage stylized features will provide feedback for subsequent style feature learning, thus guiding them to update in a dynamic and targeted manner. By learning style information from the whole domain, the style dictionary becomes more representative, which effectively overcomes the limitation of extracting style encoding from a single image. Moreover, the iterative updates of style dictionaries enhance stability and reduce uncertainty in extracting style encodings from random noise. Storing style encoding information in a multi-level dictionary helps alleviate burden on the generator and achieves effective decoupling of content features and style features. This allows it to focus on capturing domain-invariant features such as structure and shape. In this way, the multi-levels style dictionary refines complex styles within target domains at each level, ensuring that each can timely reflect latest progress in style transfer process more precise and coordinated.

### 3.3 Progressive Style Transfer Paradigm

In PSTUDA, multiple cascaded style fusion modules and decoders constitute the core components of progressive style transfer. Figure 1(b) illustrates the main processes of the first and last (decoding phase) style transfer stages, with similar style transfer processes in the intermediate stages. In the first style transfer stage, the style fusion module requires two inputs: the content feature $X_1 \in \mathbb{R}^{H \times W \times C}$ obtained by downsampling from the encoder, and the first-level style encoding $V_1 \in \mathbb{R}^{S \times S \times 1}$ from the target domain, matched with the current style fusion module. We utilize the domain label $L$ multiplied by the first style dictionary to select the target style encoding $V_1$. First, the content feature $X_1$ passes through a convolutional layer to obtain the content feature $\hat{X}_1$, and the target style encoding $V_1$ undergoes two consecutive $1 \times 1$ convolutions for channel transformation to obtain the style feature $\hat{V}_1 \in \mathbb{R}^{H \times W \times C}$.

After obtaining the content feature $\hat{X}_1$ and style feature $\hat{V}_1$, we propose a novel style fusion normalization method, Point-wise Instance Normalization (PIN), to effectively combine the two for more subtle and accurate pixel-level style transfer. The PIN can be defined as:

$$PIN(\hat{X}_l, \hat{V}_l) = \gamma_{chw}(\hat{V}_l) \cdot Norm(\hat{X}_l) + \beta_{chw}(\hat{V}_l), \tag{1}$$

$$\gamma_{chw}(\hat{V}_l), \beta_{chw}(\hat{V}_l) = Chunk(h_{(2c)hw}), \tag{2}$$

$$h_{(2c)hw} = ConvBlock(\hat{V}_l), \tag{3}$$

where $Norm(\cdot)$ denotes the Instance Normalization of the content feature $\hat{X}_l \in \mathbb{R}^{H \times W \times C}$ at layer $l$. The parameters $\gamma_{chw}$ and $\beta_{chw}$ are scaling and shifting parameters specific to the target domain. These parameters are derived by applying $ConvBlock$ and $Chunk$ operations to the style feature $\hat{V}_l \in \mathbb{R}^{H \times W \times C}$ and are adjusted to match the spatial dimensions of the content feature $\hat{X}_l$, thus enabling independent style transfer at each pixel. By rescaling the feature map using these parameters, style information specific to the target domain is integrated into the feature map for style transfer. PIN provides unique style statistics for each spatial point of the content feature, allowing for customized style fusion based on different regions and features of the image content. This facilitates finer local style variations and richer style details, making it particularly suitable for applications requiring precise style adjustments, such as medical image segmentation.

In the final stage of style transfer, the decoder takes the stylized content feature $X_n \in \mathbb{R}^{H' \times W' \times C'}$ from the previous layer and the last-level style encoding $V_n \in \mathbb{R}^{S \times S \times 1}$ as input. To spatially match the content feature $X_n$, the style encoding $V_n$ first undergoes upsampling through a deconvolutional layer for scale transformation, followed by two consecutive $1 \times 1$ convolutions for channel transformation, resulting in the style feature $\hat{V}_n \in \mathbb{R}^{H' \times W' \times C'}$. The other style fusion operations are similar to those in the first stage. There are two considerations for performing style transfer during the decoding phase. Firstly, the decoding phase is the process of image reconstruction, where integrating

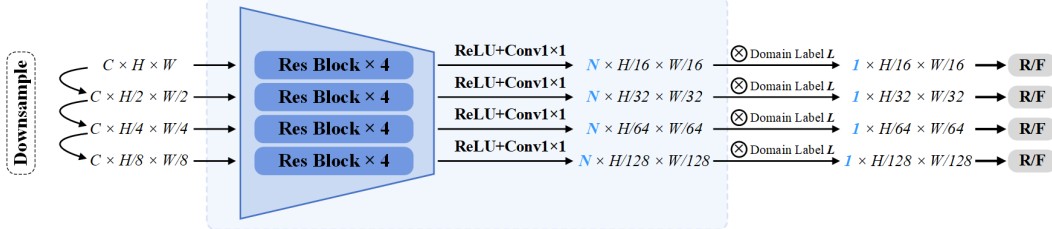

Figure 2: Architecture of the Multi-Scale Discriminator, composed of multiple residual blocks.

style features can effectively incorporate high-level style information from the target domain into the image. This helps to better express the target style while preserving content information, especially in terms of details and textures. Secondly, as the decoder is responsible for upsampling, it has the opportunity to refine and restore image details while enlarging the feature maps. Introducing style transfer at this stage ensures that these details not only conform to the content structure but also match the characteristics of the target style, thus achieving effective fusion of content and style at different scales.

### 3.4 Generator and Discriminator Architecture

**Generator**   As previously mentioned, the generated pseudo-target domain images will be used in conjunction with the corresponding segmentation annotations from the source domain to train the subsequent segmentation networks. Therefore, it is crucial to maintain structural consistency during the image translation process for the success of downstream applications. The generator in StarGAN v2, due to its multiple downsampling steps, unfortunately leads to a loss of spatial information, which poses challenges in maintaining structural consistency during image translation. In light of CycleGAN's outstanding performance in one-to-one medical image translation, we adopt the approach of OMUDA by integrating the generator architecture of CycleGAN into our encoder and decoder. To facilitate a single generator to handle one-to-multiple transfer tasks, we replace the IN layers in the decoder with PIN layers. The complete generator comprises an encoder, a decoder, and a series of cascaded style fusion modules. Among them, the encoder, equipped with two downsampling layers and multiple IN layers, is tasked with extracting domain-invariant features from the input image of the source domain. The cascaded style fusion modules are responsible for integrating the style features of the target domain with the domain-invariant features of the source domain. The decoder is composed of two upsampling layers embedded with PIN and is responsible for style transfer and image reconstruction during the decoding phase.

**Discriminator**   Inspired by Wang et al.'s work [56], we enhance the original discriminator architecture of StarGAN v2 with a multi-scale mechanism. As depicted in Fig. 2, the discriminator is composed of multiple independent discrimination branches, each containing four residual blocks [57], with each residual block having $N$ output branches serving $N$ specific target domains. The input to the discriminator includes not only the original image but also images processed through different scales of downsampling. The multi-branch output of the discriminator first multiplies with the target domain label $L$ to select the discrimination output for the corresponding domain before proceeding to authenticity determination. By evaluating the generated images at various scales, the multi-scale discriminator can more comprehensively judge the realism of the images, thereby encouraging the generator to produce images with higher quality. More detailed information on the generator and discriminator architectures can be found in Appendix B.1.

### 3.5 Loss Function

For convenience, we give the following symbol definition. $x_s \in X$ denotes a source domain image, with its corresponding source domain label $l_s \in L$. $l_t \in L$ represents any target domain label. $G$ and $D$ stand for the generator and discriminator, respectively. The loss functions in the generator include adversarial loss $\mathcal{L}_{adv}$, cycle consistency loss $\mathcal{L}_{cyc}$, and identity loss $\mathcal{L}_{idt}$ [58]. The loss in the discriminator includes adversarial loss $\mathcal{L}_{adv}$. Among them, the adversarial loss is defined as

$$\mathcal{L}_{adv} = E_{x_s, l_t} \left[ log(1 - D(G(x_s, l_t), l_t)) \right] + E_{x_s} \left[ log D(x_s, l_s) \right], \tag{4}$$

which encourages the generator to generate images that are indistinguishable from the target domain images by the discriminator. The cycle consistency loss is computed as

$$\mathcal{L}_{cyc} = E_{x_s,l_s,l_t} \left[ \| G(G(x_s, l_t), l_s) - x_s \| \right], \tag{5}$$

which ensures that an image translated from the source domain to the target domain can be translated back to the original image, thus preserving the structural consistency of the image throughout the translation process. The identity loss can be defined as

$$\mathcal{L}_{idt} = E_{x_s,l_s} \left[ \| (G(x_s, l_s) - x_s \| \right], \tag{6}$$

which is used to preserve the content of the source domain images when the generator is applied to them. The complete training loss can be summarized as follows:

$$\min_G \max_D \mathcal{L}_{adv} + \lambda_{cyc}\mathcal{L}_{cyc} + \lambda_{idt}\mathcal{L}_{idt}, \tag{7}$$

where $\lambda_{cyc}$ and $\lambda_{idt}$ denote the weights for the corresponding loss terms and they are set to 10 and 1 in our experiments, respectively.

## 4   Experiments

### 4.1   Dataset

In this study, we utilize two datasets for the performance evaluation of the method. The first one is a private dataset named MSKT, which comprises 104 cases, each including four sequences: T1c, FS T2W, T2W, and DWI. The second is the publicly available KiTS19 [59] dataset. We conduct our first set of comparative experiments using the MSKT dataset. Specifically, the annotated T1c data is served as the source domain, while the unannotated FS T2W, T2W, and DWI data are considered as the target domain. For our second set of comparative experiments, we combine KiTS19 with the MSKT dataset. In this case, all training data from KiTS19 are used as the source domain, and all four sequences of the MSKT are considered as the target domain. We ensure that there is no overlap between the source domain, target domain, and test set to prevent any potential information leakage. More details about the dataset and evaluation metrics are provided in Appendix A.

### 4.2   Comparative Study

To comprehensively evaluate the performance of the proposed method, we compare PSTUDA with five other state-of-the-art UDA methods on the MSKT and KiTS19 datasets: CycleGAN [34], MUNIT [41], SIFA [35], DEPL [39], and StarGAN v2 [22]. Except for SIFA and DEPL, the others are limited to cross-domain image transfer. Therefore, we train a dedicated U-Net [60] from scratch based on the pseudo-target domain images generated by these methods for multiple sequences. For a fair comparison, all hyperparameters in the U-Net remain consistent. Due to the one-to-one image transfer characteristics of SIFA, DEPL, CycleGAN, and MUNIT, we train a separate model for each sequence using these methods to handle multi-sequence MR image transformations.

Due to the unpaired nature of multi-sequence MR images, it is challenging to quantitatively assess the style effects and structural consistency of the generated images. Experience suggests that if anatomical structures are distorted during the image transfer process, the pseudo-target domain images generated will not correspond with the annotations of the source domain, thereby impairing segmentation performance. Moreover, if there is a significant difference between the style of the generated images and the real target domain images, this discrepancy in data distribution may also detrimentally affect the segmentation results. Hence, the quality of the generated images is indirectly indicated by the segmentation results.

**Results on MSKT**   Table 1 presents the Dice Similarity Coefficient (*DSC*) and 95% Hausdorff Distance ($HD_{95}$) results for all methods across different MR sequences. The average *DSC* indicate that our PSTUDA outperforms other methods in segmentation performance on all MRI sequences. We also observe that PSTUDA significantly surpasses MUNIT, SIFA, DEPL, and StarGAN v2 from both *DSC* and $HD_{95}$. Although SIFA and DEPL are more efficient end-to-end domain adaptation segmentation methods, they do not perform well in our task. The images generated by these methods exhibit structural distortions and lack naturalness ($4^{th}$ and $5^{th}$ columns in Fig. 3). Particularly, DEPL

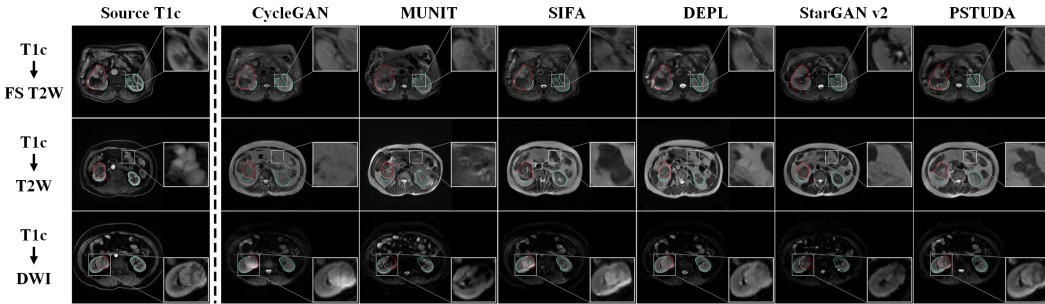

Figure 3: Qualitative results for T1c → FS T2W, T2W, and DWI on the MSKT dataset. Blue and red bounding boxes indicate the annotated boundaries of the kidney and tumor, respectively (Same below).

shows severe structural distortion of kidneys and tumors in the T2W sequence, which has greater inter-domain differences. MUNIT and StarGAN v2, while excelling in the diversity of generated images, fall short in maintaining cross-domain structural consistency, leading to distorted structures in the generated images ($3^{rd}$ and $6^{th}$ columns in Fig. 3). The performance of CycleGAN is close to that of PSTUDA on FS T2W and DWI sequences, but it underperforms on the T2W sequence, as the translated images lose many important structural details, resulting in poor image alignment. In contrast, our PSTUDA prioritizes structural consistency in the cross-domain image translation process by separating content from style and progressively recombining content and style features in a point-wise manner. This approach produces images with higher fidelity, as demonstrated in the last column of Fig. 3, thereby significantly enhancing segmentation performance.

Table 1: Quantitative segmentation results of different comparative methods on the MSKT dataset.

| Metrics | Methods | T1c→FS T2W | | | T1c→T2W | | | T1c→DWI | | |
|---|---|---|---|---|---|---|---|---|---|---|
| | | Kidney | Tumor | Avg. | Kidney | Tumor | Avg. | Kidney | Tumor | Avg. |
| *DSC* (%) | Supervised training | 90.14 | 88.73 | 89.44 | 87.53 | 80.68 | 84.11 | 90.20 | 84.76 | 87.48 |
| | W/o adaptation | 60.66 | 49.69 | 55.18 | 0.71 | 28.29 | 14.50 | 54.44 | 35.30 | 44.87 |
| | † CycleGAN [34] | **86.50** | 74.00 | 80.25 | 71.97 | 51.94 | 61.96 | **87.60** | 71.88 | 79.74 |
| | † MUNIT [41] | 80.96 | 53.00 | 66.98 | 75.78 | 48.34 | 62.06 | 78.29 | 51.87 | 65.08 |
| | † SIFA [35] | 78.31 | 50.58 | 64.45 | 65.54 | 37.10 | 51.32 | 72.94 | 43.28 | 58.11 |
| | * DEPL [39] | 85.22 | 68.20 | 76.71 | 23.44 | 23.16 | 23.30 | 83.63 | 64.26 | 73.95 |
| | † StarGAN v2 [22] | 73.58 | 41.77 | 57.68 | 42.87 | 28.68 | 35.78 | 75.26 | 41.29 | 58.28 |
| | PSTUDA(Ours) | 86.30 | **76.36** | **81.33** | **77.26** | **53.77** | **65.52** | 86.99 | **74.23** | **80.61** |
| $HD_{95}$ (mm) | Supervised training | 2.47 | 1.78 | 2.13 | 3.49 | 8.25 | 5.87 | 2.04 | 3.36 | 2.70 |
| | W/o adaptation | 11.94 | 43.03 | 27.48 | 40.14 | 69.39 | 54.77 | 11.38 | 33.60 | 22.49 |
| | † CycleGAN [34] | 4.92 | 31.54 | 18.23 | **5.37** | **6.34** | **5.86** | 4.57 | **24.62** | **14.60** |
| | † MUNIT [41] | 6.08 | 31.95 | 19.01 | 7.36 | 25.58 | 16.47 | 6.43 | 39.09 | 22.76 |
| | † SIFA [35] | 6.12 | 53.60 | 29.86 | 10.48 | 57.18 | 33.83 | 14.25 | 54.22 | 34.24 |
| | * DEPL [39] | 6.75 | 36.87 | 21.81 | 17.65 | 70.65 | 44.15 | 11.14 | 25.43 | 18.28 |
| | † StarGAN v2 [22] | 5.39 | 33.20 | 19.29 | 15.43 | 68.98 | 42.20 | 13.70 | 26.84 | 20.27 |
| | PSTUDA(Ours) | **4.66** | **25.85** | **15.25** | 6.81 | 21.37 | 14.09 | **2.96** | 34.65 | 18.80 |

† implies that we report the results of our own runs according to the official code.

* implies that the method is based on our implementation.

**Results on KiTS19 and MSKT**    This experiment utilizes CT images from KiTS19 as the source domain and four MRI sequences from MSKT as the target domain. Given the different imaging technologies employed by CT and MRI, the domain difference between them is significantly greater than that between the various MRI sequences. As shown in Table 2, most comparative methods show a reduction in kidney and tumor segmentation performance on FS T2W, T2W, and DWI when compared to the first set of experiments. Although the average *DSC* of CycleGAN on the DWI is higher than that of PSTUDA (by about 1.9%), its performance is lower than that of PSTUDA by approximately 7.5% and 9.1% on FS T2W and T2W, respectively. Our PSTUDA almost achieves the

best average *DSC* results on all sequences. Compared to the first set of experiments, the performance of CycleGAN has decreased by about 6.8% and 4.4% on FS T2W and T2W, respectively, while the performance of the proposed PSTUDA has decreased by 0.5% and increased by 1.2% on these two sequences. These results underscore PSTUDA's exceptional robustness in tasks with larger domain differences. As shown in Fig. 4 in Appendix C, our PSTUDA outperforms other comparative methods in maintaining structural consistency in translated images. Moreover, thanks to the one-to-multiple features, our method also shows a significant advantage in training efficiency, which will be detailed in the subsequent section. In summary, the highlight of PSTUDA lies in its dual improvement of segmentation performance and training efficiency.

Table 2: Quantitative segmentation results of different comparative methods on the KiTS19 and MSKT datasets.

| Metrics | Methods | CT→T1c | | | CT→FS T2W | | | CT→T2W | | | CT→DWI | | |
|---|---|---|---|---|---|---|---|---|---|---|---|---|---|
| | | Kidney | Tumor | Avg. | Kidney | Tumor | Avg. | Kidney | Tumor | Avg. | Kidney | Tumor | Avg. |
| *DSC* (%) | Supervised training | 90.74 | 85.69 | 88.22 | 90.14 | 88.73 | 89.44 | 87.53 | 80.68 | 84.11 | 90.20 | 84.76 | 87.48 |
| | W/o adaptation | 71.20 | 13.27 | 42.24 | 43.75 | 6.27 | 25.01 | 9.13 | 22.31 | 15.72 | 49.25 | 4.13 | 26.69 |
| | † CycleGAN [34] | **85.96** | 70.63 | 78.30 | 79.08 | 67.64 | 73.36 | 69.68 | 45.52 | 57.60 | **87.45** | **81.35** | **84.40** |
| | † MUNIT [41] | 76.71 | 67.21 | 71.96 | 72.36 | 66.70 | 69.53 | 72.55 | 47.52 | 60.04 | 75.37 | 55.32 | 65.35 |
| | † SIFA [35] | 76.34 | 48.12 | 62.23 | 71.69 | 56.56 | 64.13 | 49.39 | 23.25 | 36.32 | 72.31 | 31.41 | 51.86 |
| | * DEPL [39] | 82.57 | 73.08 | 77.83 | 81.31 | 71.87 | 76.59 | 37.95 | 28.77 | 33.36 | 82.70 | 71.51 | 77.11 |
| | † StarGAN v2 [22] | 53.43 | 24.82 | 39.13 | 62.96 | 21.53 | 42.25 | 47.51 | 13.13 | 30.32 | 69.36 | 10.89 | 40.13 |
| | PSTUDA(Ours) | 84.85 | **73.44** | **79.15** | 83.99 | **77.70** | **80.85** | 76.56 | **56.85** | **66.71** | 85.91 | 79.17 | 82.54 |
| $HD_{95}$ (mm) | Supervised training | 5.98 | 24.24 | 15.11 | 2.47 | 1.78 | 2.13 | 3.49 | 8.25 | 5.87 | 2.04 | 3.36 | 2.70 |
| | W/o adaptation | 11.51 | 56.79 | 34.15 | 20.75 | 64.69 | 42.72 | 45.22 | 100 | 72.61 | 17.23 | 73.67 | 45.45 |
| | † CycleGAN [34] | **6.57** | 32.86 | 19.72 | 5.22 | **4.12** | **4.67** | 6.33 | 29.38 | 17.85 | 5.48 | **2.78** | **4.13** |
| | † MUNIT [41] | 7.66 | 53.06 | 30.36 | 4.96 | 16.94 | 10.95 | 6.40 | 32.34 | 19.37 | **4.43** | 19.12 | 11.78 |
| | † SIFA [35] | 23.06 | 77.79 | 50.42 | 6.15 | 28.08 | 17.12 | 11.08 | 87.59 | 49.33 | 10.16 | 59.71 | 34.93 |
| | * DEPL [39] | 16.91 | 50.66 | 33.78 | 5.44 | 36.37 | 20.90 | 14.91 | 70.16 | 42.54 | 6.04 | 23.34 | 14.69 |
| | † StarGAN v2 [22] | 15.95 | 65.61 | 40.78 | 7.15 | 50.28 | 28.71 | 19.07 | 78.08 | 48.57 | 24.07 | 61.15 | 42.61 |
| | PSTUDA(Ours) | 6.90 | **18.97** | **12.94** | **3.99** | 26.17 | 15.08 | 7.54 | **20.08** | **13.81** | 5.05 | 27.64 | 16.35 |

**Training efficiency**    In this section, we evaluate the performance of PSTUDA in comparison with CycleGAN, MUNIT, and StarGAN v2 in terms of model parameters and FLOPs. As shown in Table 9 in Appendix D, the FLOPs for CycleGAN and MUNIT represent the cumulative results of the image translation tasks for the three domains, *i.e.*, T1c→FS T2W, T1c→T2W, and T1c→DWI. For the transfer of T1c to the other three MR sequences using CycleGAN, MUNIT, and StarGAN v2, the parameters that need to be optimized are 2, 3.3, and 1.9 times that of PSTUDA, respectively. Regarding FLOPs, PSTUDA demonstrates a reduction of 72% and 79% compared to CycleGAN and MUNIT, respectively, but shows a 17% increase compared to StarGAN v2. The increased computational demand is mainly due to the PIN module in PSTUDA, which requires per-pixel fusion of style and content features. It is worth noting that the training efficiency of PSTUDA becomes more advantageous when more target domains are included in a one-to-multiple domain adaptation task. Overall, PSTUDA substantially decreases both model parameters and FLOPs while maintaining high segmentation accuracy, implying lower memory requirements and faster inference speed. In summary, these results highlight the significant advantages of PSTUDA in enhancing the efficiency and feasibility of clinical applications.

## 4.3   Ablation Study

**Effectiveness of PST and MS_D**    To investigate the effectiveness of the proposed Progressive Style Transfer (PST) paradigm, we conduct ablation studies on it with the Multi-Scale Discriminator (MS_D) on the MSKT dataset. As shown in Table 3 , the introduction of PST significantly improves the *DSC* and $HD_{95}$ metrics compared to the Baseline. This reflects the ability of the PST to enhance the realism of translated images while maintaining structural consistency. Furthermore, the multi-scale discriminator assesses the pseudo-target domain images at different scales, thereby further enhancing the capabilities of the generator.

**Effectiveness of M_SD and PIN**    We also conduct ablation studies on the internal sub-modules within PST, namely the Multi-level Style Dictionary (M_SD) and the Point-wise Instance Normalization (PIN). As illustrated in Table 4 , replacing the Mapping Network and Style Encoder in the

Baseline with the M_SD results in a significant performance improvement. This can be attributed to the stability and representativeness of the learned style encodings. Moreover, the integration of the M_SD with PIN fully considers the details and local style variations of images, thereby yielding exceptional results.

Table 3: Ablation study of Progressive Style Transfer paradigm (PST) and Multi-Scale Discriminator (MS_D) on the MSKT dataset. The Baseline model is StarGAN v2.

| Modules | | Metrics | T1c→FS T2W | | | T1c→T2W | | | T1c→DWI | | |
|---|---|---|---|---|---|---|---|---|---|---|---|
| PST | MS_D | | Kidney | Tumor | Avg. | Kidney | Tumor | Avg. | Kidney | Tumor | Avg. |
| | | | 74.96 | 58.46 | 66.71 | 58.48 | 17.05 | 37.77 | 69.29 | 41.41 | 55.35 |
| ✓ | | $DSC$ (%) | 84.94 | **77.74** | **81.34** | 70.16 | 52.80 | 61.48 | 85.09 | 72.06 | 78.58 |
| ✓ | ✓ | | **86.30** | 76.36 | 81.33 | **77.26** | **53.77** | **65.52** | **86.99** | **74.23** | **80.61** |
| | | | 5.33 | 59.68 | 32.50 | 11.44 | 93.41 | 52.42 | 14.55 | 68.95 | 41.75 |
| ✓ | | $HD_{95}$ (mm) | 5.70 | 33.03 | 19.36 | 8.45 | 52.71 | 30.58 | 4.22 | **10.04** | **7.13** |
| ✓ | ✓ | | **4.66** | **25.85** | **15.25** | **6.81** | **21.37** | **14.09** | **2.96** | 34.65 | 18.80 |

Table 4: Ablation study of Multi-level Style Dictionary (M_SD) and Point-wise Instance Normalization (PIN) in PST on the MSKT Dataset. The Baseline model is StarGAN v2.

| Modules | | Metrics | T1c→FS T2W | | | T1c→T2W | | | T1c→DWI | | |
|---|---|---|---|---|---|---|---|---|---|---|---|
| M_SD | PIN | | Kidney | Tumor | Avg. | Kidney | Tumor | Avg. | Kidney | Tumor | Avg. |
| | | | 74.96 | 58.46 | 66.71 | 58.48 | 17.05 | 37.77 | 69.29 | 41.41 | 55.35 |
| ✓ | | $DSC$ (%) | 83.75 | 71.48 | 77.62 | 65.82 | 30.45 | 48.14 | 77.38 | 54.99 | 66.19 |
| ✓ | ✓ | | **84.94** | **77.74** | **81.34** | **70.16** | **52.80** | **61.48** | **85.09** | **72.06** | **78.58** |
| | | | 5.33 | 59.68 | 32.50 | 11.44 | 93.41 | 52.42 | 14.55 | 68.95 | 41.75 |
| ✓ | | $HD_{95}$ (mm) | **4.89** | 40.75 | 22.82 | 8.94 | 53.27 | 31.11 | 4.50 | 59.67 | 32.08 |
| ✓ | ✓ | | 5.70 | **33.03** | **19.36** | **8.45** | **52.71** | **30.58** | **4.22** | **10.04** | **7.13** |

**Ablation of PIN and other normalization methods** We compare PIN with AdaIN [52] and BIN [53], and as shown in Table 5, PIN outperforms the others. We attribute this superior performance to PIN's ability to account for local style differences, which is particularly advantageous for fine-grained segmentation tasks (*e.g.*, kidney tumors, as abnormal pathological tissues, exhibit significant style differences). This capability enables PSTUDA to generate synthetic images that align more closely with the data distribution of the target domain.

Table 5: Ablation study of PIN with AdaIN and BIN on the MSKT dataset.

| Normalization | Metrics | T1c→FS T2W | | | T1c→T2W | | | T1c→DWI | | |
|---|---|---|---|---|---|---|---|---|---|---|
| | | Kidney | Tumor | Avg. | Kidney | Tumor | Avg. | Kidney | Tumor | Avg. |
| AdaIN | | 85.05 | 62.11 | 73.58 | 75.40 | 43.32 | 59.36 | 83.69 | 64.44 | 74.07 |
| BIN | $DSC$ (%) | 82.32 | 67.91 | 75.12 | 74.14 | 49.02 | 61.58 | 85.85 | 65.07 | 75.46 |
| PIN | | **86.30** | **76.36** | **81.33** | **77.26** | **53.77** | **65.52** | **86.99** | **74.23** | **80.61** |
| AdaIN | | 5.46 | **24.30** | **14.88** | **6.15** | 33.44 | 19.79 | 5.22 | 44.20 | 24.71 |
| BIN | $HD_{95}$ (mm) | 9.71 | 53.73 | 31.72 | 7.05 | 36.79 | 21.92 | 5.36 | 61.74 | 33.55 |
| PIN | | **4.66** | 25.85 | 15.26 | 6.81 | **21.37** | **14.09** | **2.96** | **34.65** | **18.81** |

# 5 Conclusion

In this work, we propose a novel and efficient *One-to-Multiple* PSTUDA framework that utilizes a multi-level style dictionary to decouple and store style information. By employing multiple cascaded style fusion modules, our framework progressively recombines content and style features, thereby achieving superior cross-modal alignment and consistency of medical tissue structures. Experimental validation on both a private dataset and a public dataset demonstrates the significant advantages of our method in improving training efficiency for one-to-multiple domain adaptation tasks and enhancing the accuracy of multi-sequence kidney tumor segmentation.

## Acknowledgments

This work was supported in part by the National Natural Science Foundation of China under Grants 62272404, 62076007, and 62372170, in part by the Natural Science Foundation of Hunan Province of China under Grants 2022JJ30571 and 2023JJ40638, and in part by the Research Foundation of Education Department of Hunan Province of China under Grant 23A0146.

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

# Appendix

In the appendix of this paper, Section A provides a detailed introduction to the experimental datasets and evaluation metrics. Subsequently, Section B offers the specific implementation details of the experiments. Section C presents a visual comparison of the experimental results from different methods on the KiTS19 and MSKT datasets. Section D shows a comparison of training efficiency among different methods. Sections E and F present two sets of supplementary comparative experiments and two sets of supplementary ablation experiments, respectively. Finally, Section G discusses the potential limitations and the broad impacts of our method.

## A    Details of the Experimental Datasets and Evaluation Metrics

**MSKT**    The MSKT dataset contains 104 cases of multi-sequence kidney and kidney tumor MRI data. Each case includes four MRI sequences: T1c, FS T2W, T2W, and DWI. The data are stored in NRRD format and scanned in axial view. The MR images have a resolution of $512 \times 512 \times N$, where $N$ represents the number of slices, ranging from 11 to 96. The slice thickness varies from 2 to 6 millimeters. Annotations were created by experienced clinicians, with normal kidney tissue regions labeled as 1, kidney tumor regions as 2, and other areas labeled as 0, considered as the background. In the first set of comparative experiments (Section 4.2), we randomly divide all cases into source domain training sets, target domain training sets, and target domain test sets in a 4.5:4.5:1 ratio. The annotated T1c data, comprising 46 cases, serve as the source domain, while the unannotated FS T2W, T2W, and DWI data, each comprising 46 cases, constitute the target domain. The remaining 12 cases are used as the test set. This division ensures that there is no intersection between the source domain, target domain, and test set to prevent information leakage. The number of samples and slices included in each sequence is detailed in Table 6.

To protect patient privacy, all data were anonymized during collection and processing, removing any information that could identify the patients. The use of this dataset has been reviewed and approved by the Institutional Review Board (IRB) to ensure that the research complies with ethical standards and safeguards participant rights. We strictly adhere to relevant laws and regulations regarding data use and protection to ensure the legal and compliant use of the data.

**KiTS19**    The KiTS19 dataset, established for the 2019 kidney and kidney tumor segmentation challenge, comprises multi-phase CT scans along with corresponding segmentation annotations for 300 patients. A total of 210 cases are designated for model training, and the remaining 90 are used to evaluate model performance. The CT images have a resolution of $512 \times 512 \times N$, where $N$ ranges from 29 to 1,059 slices. For our second set of comparative experiments (Section 4.2), we combine the KiTS19 dataset with the MSKT dataset. Specifically, as the KiTS19 challenge initially provides only the training set, we use all cases from this set as our source domain training set, which includes 45,424 CT slices, with 16,336 slices containing targets. All four sequences of the MSKT are considered as the target domain, and we merge the 92 cases identified as source and target domains in Table 6 to form the target domain training set, while the test set remains unchanged. Please note that in all of our experiments, the slices used for training all contain the kidney or tumor targets, while the slices used for testing are all the slices in each case.

**Evaluation metrics**    Following [21], the Dice Similarity Coefficient (*DSC*) and the 95% Hausdorff Distance (*HD*$_{95}$) are utilized for quantitative comparisons. It is important to note that if the predicted results and the true annotations do not contain the same categories, directly calculating *HD*$_{95}$ may lead to errors. To address this issue, we set the *HD*$_{95}$ value to 100 in cases of potential calculation errors, indicating no overlap between the predicted results and the true annotations.

## B    Implementation Details

All experiments are conducted on an NVIDIA A800 GPU utilizing the PyTorch framework. CT and MR data are uniformly processed to a resolution of $256 \times 256$. In the one-to-multiple domain adaptation stage, the number of domains is set to 4 and 5 for the two sets of experiments, respectively (including source and target domains). The length of the multi-level style dictionary is set to 4,096. The number of style fusion modules in the generator is 4. The weights of adversarial loss, cycle

Table 6: Data partitioning for each sequence in the MSKT dataset.

| Sequence | Source | | | Target | | | Test | | |
|---|---|---|---|---|---|---|---|---|---|
| | cases | slices | slices with target | cases | slices | slices with target | cases | slices | slices with target |
| T1c | 46 | 3444 | 2196 | - | - | - | - | - | - |
| FS T2W | - | - | - | 46 | 1110 | 802 | 12 | 288 | 219 |
| T2W | - | - | - | 46 | 1097 | 754 | 12 | 288 | 204 |
| DWI | - | - | - | 46 | 1102 | 783 | 12 | 288 | 213 |

consistency loss, and identity loss are set to 1, 10, and 1, respectively. Network optimization employs the Adam optimizer [61] with an initial learning rate of 1e-4, and the batch size is 8. Other hyperparameters follow the configuration in StarGAN v2. In the segmentation stage, U-Net is chosen as the network architecture, paired with the Adam optimizer starting with a learning rate of 1e-4. The batch size is set to 16, with a total training duration of 50 epochs. In the above setup, it takes about 18 hours for our method to fully execute once.

## B.1 Network Architecture

**Generator**    As shown in Table 7, the generator within PSTUDA comprises an encoder, four style fusion modules, and a decoder. The encoder includes three sets of convolutional blocks and four Resnet blocks, while the decoder is composed of two upsampling layers with transposed convolution operations and one convolutional block. Instance Normalization is applied during the encoding phase, and Point-wise Instance Normalization is utilized within the style fusion modules and the decoder.

Table 7: Architecture of the generator.

| Layer | Stride | Padding | Norm | Repeat | Output shape |
|---|---|---|---|---|---|
| Input $x$ | - | - | - | - | $256 \times 256 \times 1$ |
| Encoder: Conv7$\times$7 | 1 | 0 | IN | 1 | $256 \times 256 \times 64$ |
| Encoder: Conv3$\times$3 | 2 | 1 | IN | 1 | $128 \times 128 \times 128$ |
| Encoder: Conv3$\times$3 | 2 | 1 | IN | 1 | $64 \times 64 \times 256$ |
| Encoder: Resnet Block | 1 | 0 | IN | 4 | $64 \times 64 \times 256$ |
| Style Fusion Module | 1 | 0 | PIN | 4 | $64 \times 64 \times 256$ |
| Decoder: upsample | 2 | 1 | PIN | 1 | $128 \times 128 \times 128$ |
| Decoder: upsample | 2 | 1 | PIN | 1 | $256 \times 256 \times 64$ |
| Decoder: Conv7$\times$7 | 1 | 0 | - | 1 | $256 \times 256 \times 1$ |

**Discriminator**    As shown in Fig. 2 in Section 3.4, we utilize four independent and identical discrimination branches in the multi-scale discriminator. The detailed architecture of the first discrimination branch, which takes the original size image as input, is presented in Table 8.

Table 8: Architecture of the discriminator.

| Layer | Stride | Padding | Repeat | Output shape |
|---|---|---|---|---|
| Input $x$ | - | - | - | $256 \times 256 \times 1$ |
| Conv3$\times$3 | 1 | 1 | 1 | $256 \times 256 \times 64$ |
| ResBlk | 1 | 1 | 1 | $128 \times 128 \times 128$ |
| ResBlk | 1 | 1 | 1 | $64 \times 64 \times 256$ |
| ResBlk | 1 | 1 | 1 | $32 \times 32 \times 512$ |
| ResBlk | 1 | 1 | 1 | $16 \times 16 \times 512$ |
| Conv1$\times$1 | 1 | 0 | 1 | $16 \times 16 \times 512$ |
| Conv1$\times$1 | 1 | 0 | 1 | $16 \times 16 \times 4$ |

## B.2 Implementation of Comparative Methods

**CycleGAN**   This method is derived from the GitHub repository: https://github.com/junyanz/pytorch-CycleGAN-and-pix2pix. In our experiments, ResNet_9block and PatchGAN with 3 convolutional layers are selected as the generator and discriminator for CycleGAN, respectively. The weights for adversarial loss (MSE), cycle consistency loss (L1), and domain-invariant perceptual loss (L1) are set to 1, 10, and 5, respectively. Other hyperparameters are consistent with the open-source code. During the segmentation phase, all hyperparameters for U-Net are the same as those in our PSTUDA.

**MUNIT**   This method is from the GitHub repository: https://github.com/NVlabs/MUNIT. The architecture of the MUNIT network remains the same as the open-source code. The weights for adversarial loss (MSE), image reconstruction loss (L1), style reconstruction loss (L1), and content reconstruction loss (L1) are set to 1, 10, 1, and 1, respectively. In the segmentation stage, all U-Net hyperparameters are identical to those in our PSTUDA.

**SIFA**   This method is available from the GitHub repository: https://github.com/JianghaoWu/SIFA-pytorch.   The official version of this method is implemented in TensorFlow, available at https://github.com/cchen-cc/SIFA. In OMUDA, they have re-implemented SIFA in PyTorch, and we have used the PyTorch version of SIFA to ensure all methods use the same framework for a fair comparison. All settings, hyperparameters, and losses used in our experiments are the same as those in the open-source code.

**DEPL**   This method is based on our own implementation.

**StarGAN v2**   This method is from the GitHub repository: https://github.com/clovaai/stargan-v2. In our experiments, the network architecture is consistent with the official code. The weights for R1 regression loss, adversarial loss (CE), cycle consistency loss (L1), style reconstruction loss (L1), and diversity-sensitive loss (L1) are all set to 1. High-pass filtering is not used in this comparative experiment. During the segmentation phase, all U-Net hyperparameters are the same as those in PSTUDA.

## C   Visualization of the KiTS19 and MSKT Experiments

In this section, we present the visualized results of the second set of comparative experiments conducted on the KiT19 and MSKT datasets, and these visualizations are discussed in Section 4.2.

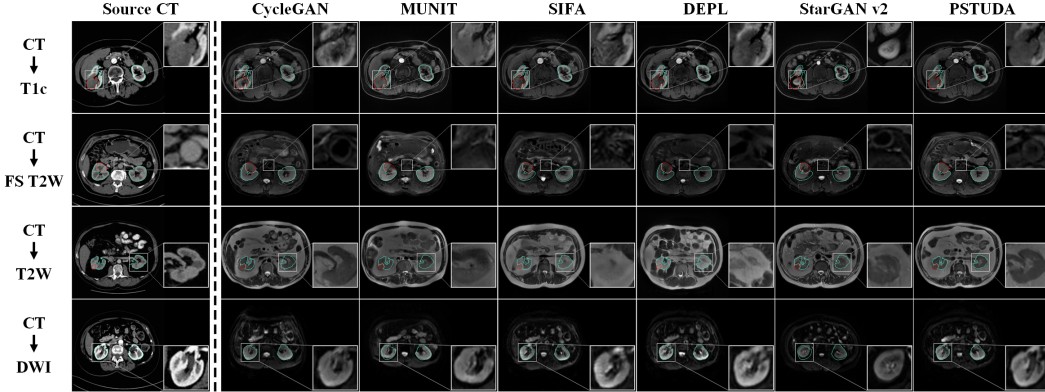

Figure 4: Qualitative results for CT $\rightarrow$ T1c, FS T2W, T2W, and DWI on the KiTS19 and MSKT datasets.

## D    Training Efficiency

This section displays the comparative outcomes in terms of model parameters and FLOPs among different cross-domain image translation methods. As illustrated in Table 9, we compare PSTUDA with CycleGAN, MUNIT, and StarGAN v2, because these methods do not involve a segmentation stage. It is worth note that the style dictionary itself does not contribute to the computation of FLOPs. All models are trained using FP32 precision. The detailed analysis of these comparative results is provided in Section 4.2.

Table 9: Model parameters and FLOPs of different methods.

| Methods | Components | Parameters | FLOPs |
|---------|-----------|-----------|-------|
| CycleGAN | $G_A$ | $11.37M \times 3$ | $112.08G \times 3$ |
| | $G_B$ | $11.37M \times 3$ | $112.08G \times 3$ |
| | $D_A$ | $2.76M \times 3$ | $6.24G \times 3$ |
| | $D_B$ | $2.76M \times 3$ | $6.24G \times 3$ |
| | Total | $28.26M \times 3$ | $236.64G \times 3$ |
| MUNIT | $G_A$ | $15.01M \times 3$ | $152.18G \times 3$ |
| | $G_B$ | $15.01M \times 3$ | $152.18G \times 3$ |
| | $D_A$ | $8.27M \times 3$ | $4.28G \times 3$ |
| | $D_B$ | $8.27M \times 3$ | $4.28G \times 3$ |
| | Total | $46.56M \times 3$ | $312.92G \times 3$ |
| StarGAN v2 | $G$ | $33.89M$ | $97.30G$ |
| | $N_M$ | $4.08M$ | $0.01G$ |
| | $E_S$ | $20.98M$ | $32.98G$ |
| | $D$ | $20.85M$ | $32.98G$ |
| | Total | $79.80M$ | $163.27G$ |
| PSTUDA | $G$ | $12.17M$ | $154.84G$ |
| | $D$ | $29.92M$ | $41.46G$ |
| | $D_S$ | $0.1M$ | - |
| | Total | $42.19M$ | $196.30G$ |

$G$: Generator, $D$: Discriminator, $N_M$: Mapping Network, $E_S$: Style Encoder, $D_S$: Style Dictionary.

## E    Supplementary Comparative Experiments

To further validate the generalization capability of PSTUDA, we conduct bidirectional cross-modal validation experiments on a publicly available abdominal multi-organ dataset [62] and perform reverse validation experiments from MR to CT on the MSKT and KiTS19 datasets. The results, as shown in Tables 10 and 11, indicate that our method significantly outperforms StarGAN v2 (baseline) in both experimental groups.

## F    Supplementary Ablation Experiments

The multi-level style dictionary and style fusion module are the core components of PSTUDA, and these two modules include the following hyperparameters: the depth of the style dictionary, the number of levels in the style dictionary, and the number of style fusion modules. It is important to note that the number of levels in the style dictionary is equal to the number of style fusion modules. To investigate the sensitivity of model performance to various hyperparameters, we conduct extensive ablation studies on the depth of the multi-level style dictionary (M_SD) and the number of style fusion modules (SFM). As presented in Tables 12 and 13, under the original settings (dictionary

Table 10: Quantitative segmentation results for bidirectional cross-modal experiments on the abdominal multi-organ dataset.

| Metrics | Methods | CT→MR | | | | | MR→CT | | | | |
|---|---|---|---|---|---|---|---|---|---|---|---|
| | | Liver | R. kidney | L. kidney | Spleen | Avg. | Liver | R. kidney | L. kidney | Spleen | Avg. |
| DSC (%) | Supervised training | 94.39 | 90.86 | 73.38 | 78.00 | 84.16 | 87.45 | 69.33 | 77.76 | 75.61 | 77.54 |
| | W/o adaptation | 23.44 | 1.99 | 12.77 | 20.28 | 14.62 | 37.33 | 0.00 | 0.26 | 1.29 | 9.72 |
| | StarGAN v2 | 75.04 | 68.16 | 62.83 | 68.13 | 68.54 | 44.32 | 28.05 | 26.51 | 24.85 | 30.93 |
| | PSTUDA | **88.15** | **74.20** | **70.51** | **71.99** | **76.21** | **53.54** | **53.42** | **60.28** | **38.48** | **51.43** |
| HD$_{95}$ (mm) | Supervised training | 2.04 | 51.59 | 5.86 | 84.09 | 35.89 | 14.72 | 15.68 | 38.17 | 54.15 | 30.68 |
| | W/o adaptation | 37.91 | 75.82 | 62.57 | 75.80 | 63.03 | 33.22 | 80.59 | 49.00 | 65.36 | 57.04 |
| | StarGAN v2 | 45.12 | 57.08 | **61.45** | 115.49 | 69.78 | **33.70** | 71.15 | 75.29 | 100.16 | 70.08 |
| | PSTUDA | **21.49** | **48.35** | 70.11 | **46.24** | **46.55** | 43.02 | **46.12** | **12.13** | **54.15** | **38.85** |

Table 11: Quantitative segmentation results from MR to CT on the MSKT and KiTS19 datasets.

| Metrics | Methods | MR(T1c)→CT | | | MR(FS T2W)→CT | | |
|---|---|---|---|---|---|---|---|
| | | Kidney | Tumor | Avg. | Kidney | Tumor | Avg. |
| DSC (%) | Supervised training | 91.98 | 66.61 | 79.30 | 91.98 | 66.61 | 79.30 |
| | W/o adaptation | 41.87 | 16.26 | 29.07 | 8.07 | 11.36 | 9.72 |
| | StarGAN v2 | 34.44 | 24.02 | 29.23 | 46.17 | 22.96 | 34.57 |
| | PSTUDA | **73.13** | **56.23** | **64.68** | **78.13** | **50.85** | **64.49** |
| HD$_{95}$ (mm) | Supervised training | 33.66 | 45.89 | 39.78 | 33.66 | 45.89 | 39.78 |
| | W/o adaptation | 64.91 | 100.61 | 82.76 | 57.90 | 102.54 | 80.22 |
| | StarGAN v2 | 69.72 | 96.61 | 83.17 | 57.99 | 92.36 | 75.17 |
| | PSTUDA | **56.95** | **93.65** | **75.30** | **40.82** | **50.25** | **45.54** |

Table 12: Ablation study on the depth of the Multi-level Style Dictionary (M_SD) on the MSKT dataset.

| M_SD Depth | Metrics | T1c→FS T2W | | | T1c→T2W | | | T1c→DWI | | |
|---|---|---|---|---|---|---|---|---|---|---|
| | | Kidney | Tumor | Avg. | Kidney | Tumor | Avg. | Kidney | Tumor | Avg. |
| 256 | | 84.85 | 66.34 | 75.60 | 70.55 | 43.85 | 57.20 | 83.15 | 63.52 | 73.34 |
| 1024 | DSC (%) | 85.84 | 69.68 | 77.76 | 70.06 | 42.76 | 56.41 | 83.99 | 64.41 | 74.20 |
| 4096 | | **86.30** | **76.36** | **81.33** | 77.26 | 53.77 | 65.52 | **86.99** | **74.23** | **80.61** |
| 16384 | | 84.63 | 66.74 | 75.69 | **77.63** | **57.49** | **67.56** | 83.31 | 68.78 | 76.05 |

Table 13: Ablation study on the number of Style Fusion Modules (SFM) on the MSKT dataset.

| SFM Number | Metrics | T1c→FS T2W | | | T1c→T2W | | | T1c→DWI | | |
|---|---|---|---|---|---|---|---|---|---|---|
| | | Kidney | Tumor | Avg. | Kidney | Tumor | Avg. | Kidney | Tumor | Avg. |
| 2 | | 85.65 | 71.77 | 78.71 | 73.94 | 44.51 | 59.23 | 83.89 | 61.31 | 72.60 |
| 4 | DSC (%) | **86.30** | **76.36** | **81.33** | **77.26** | **53.77** | **65.52** | **86.99** | **74.23** | **80.61** |
| 6 | | 84.49 | 67.94 | 76.22 | 74.97 | 50.57 | 62.77 | 81.20 | 60.44 | 70.82 |
| 8 | | 84.09 | 63.86 | 73.98 | 60.06 | 30.43 | 45.25 | 83.75 | 60.60 | 72.18 |

depth of 4,096 and module number of 4), PSTUDA performs optimally in most experiments. Notably, the segmentation performance on the T2W sequence improves when the dictionary depth is increased to 16,384.

# G   Limitations and Broader Impacts

**Limitations**   Our work presents a domain adaptation method for multi-target domain image translation. However, the field of one-to-multiple domain adaptation is not extensively explored, and our

study may have some limitations that should be acknowledged. Firstly, the tasks we focus on are confined to domain adaptation of multi-sequence MR images and the segmentation of kidneys and tumors as presented in this work. In future research, we aim to broaden our scope to include domain adaptation for other medical and natural images, as well as other downstream tasks such as object detection. Secondly, compared to the ideal one-to-multiple model, our current model inherently operates within a broader multi-domain to multi-domain adaptation framework. Therefore, in our future studies, we aspire to design a more specialized one-to-multiple model, which is expected to further improve model performance and reduce model complexity.

**Broader impacts**  In fact, the impact of one-to-multiple domain adaptation technology is profound, especially when facing numerous target domains and limited annotated data. Our PSTUDA framework can efficiently extend an annotated source domain to adapt to multiple unannotated new domains, significantly reducing the time and resource investment required for multi-domain transfer tasks. This capability is crucial in complex medical data environments, as it not only greatly alleviates the workload of medical professionals but also holds promise for improving the diagnosis and treatment processes for patients. Overall, our research enhances the flexibility and efficiency of domain adaptation technology, which advances the field of machine learning and brings innovative solutions to critical industries such as healthcare.

