# OpenReview forum: "One-to-Multiple: A Progressive Style Transfer Unsupervised Domain-Adaptive Framework for Kidney Tumor Segmentation"
_NeurIPS.cc/2024/Conference — NeurIPS 2024 poster_

### Official Review · Reviewer_pquR · 2024-07-08

**Soundness:** 3
**Presentation:** 3
**Contribution:** 3
**Rating:** 8
**Confidence:** 5

**Summary:**

The paper proposes the One-to-Multiple Progressive Style Transfer Unsupervised Domain-Adaptive (PSTUDA) framework for kidney and tumor segmentation in multi-sequence MRI, addressing inefficiencies in existing one-to-one UDA methods. PSTUDA features a multi-level style dictionary and multiple cascading style fusion modules using point-wise instance normalization to recombine content and style features progressively. Tested on private (MSKT) and public (KiTS19) datasets, PSTUDA showed 1.8% and 3.9% improvements in Dice Similarity Coefficient for kidney and tumor segmentation, respectively, and achieved significant reductions in model parameters and computational costs. The framework outperformed state-of-the-art UDA methods, enhancing both segmentation performance and training efficiency.

**Strengths:**

This paper demonstrates several strengths across various dimensions.

In terms of originality, PSTUDA introduces a relatively interesting approach by employing cascaded style fusion modules and Point-wise Instance Normalization (PIN) to achieve excellent cross-modal alignment and structural consistency. This combo allows for more precise and progressive recombination of content and style features, addressing limitations in existing one-to-one UDA methods and extending the applicability of domain adaptation techniques in medical imaging.

The strong review of related works situates PSTUDA within the broader context of domain adaptation and medical image segmentation.

The paper is well-written and clear, providing detailed explanations of the architecture and methodologies. It emphasizes structural consistency, which is critical for medical image analysis. The comprehensive writing extends to the appendices, offering an appropriate level of detail on datasets, which aids in reproducibility and understanding

PSTUDA’s experimental design is robust, featuring both comparative and ablation studies as well as appropriate metrics and analysis.

This paper is original, detailed, clear, and demonstrates reasonable improvements in segmentation with significant improvements in floating point operations and model size.

**Weaknesses:**

- The reported improvements in Dice Similarity Coefficient (DSC) and 95% Hausdorff Distance (HD95), while notable, are relatively modest (1.8% and 3.9%, respectively).

- The framework's scalability to handle a larger number of target domains in real-world scenarios remains untested.

- The generalizability of this approach beyond simple kidney/tumour segmentation into multi-class intra-organ segmentation or even multi-organ segmentation remains to be seen.

**Questions:**

- I would agree with the concept of the multiple cascading modules and Point-wise Instance Normalization but am surprised it doesn't benefit the overall segmentation results further. Can the authors provide their insights into why there isn't a more dramatic boost? What future work do they envision to achieve more substantial performance boosts?

-  paper demonstrates the effectiveness of PSTUDA on kidney and tumor segmentation in multi-sequence MRI. Have the authors tested the framework on other medical imaging tasks or modalities, such as CT or ultrasound? What would expected results and challenges be in other modalities?

- Despite the reduction in model parameters and FLOPs, PSTUDA still appears resource-intensive. How feasible is the deployment of PSTUDA in real-world clinical settings with limited computational infrastructure?

- Are there any plans to release code, pre-trained models, or detailed implementation guides? How can the research community best leverage PSTUDA in their own work?

**Limitations:**

Limitations and Broader Impacts are appropriately addressed.

---

> ### Author Rebuttal · Authors · 2024-08-07
>
> We would like to thank you for the positive consideration and the useful feedback on our work. We address all your concerns below.
>
> **W1. About Performance Enhancement.**
>
> The one-to-multiple UDA task presents many additional challenges compared to the one-to-one task. For example, the differences between various target domains are distinct, and a single generator must coordinate the complex mapping relationships among multiple target domains, thereby making the learning process more difficult. However, a single generator can translate one source domain into multiple target domains, significantly reducing the training time and resource costs compared to one-to-one methods. Overall, our findings have shown significant advantages and we believe our approach holds substantial benefits for multi-sequence UDA tasks.
>
> **W2. Scalability to more target domains.**
>
> As you suggested, we have acquired a new set of T1 MRI sequences from our partner hospital to validate the performance of our method in a one-to-five domain adaptive segmentation task. As shown in the table below, our method maintains high performance even with an increased number of target domains, which are derived from real-world clinical scenarios.
> |     **Methods**     |           | **CT→T1c** |           |           | **T1c→FS T2W** |           |           | **CT→T2W** |           |           | **CT→DWI** |           |           | **CT→T1** |           |
> | :-----------------: | :-------: | :--------: | :-------: | :-------: | :------------: | :-------: | :-------: | :--------: | :-------: | :-------: | :--------: | :-------: | :-------: | :-------: | :-------: |
> |                     |  Kidney   |   Tumor    |   Avg.    |  Kidney   |     Tumor      |   Avg.    |  Kidney   |   Tumor    |   Avg.    |  Kidney   |   Tumor    |   Avg.    |  Kidney   |   Tumor   |   Avg.    |
> | Supervised training |   90.74   |   85.69    |   88.22   |   90.14   |     88.73      |   89.44   |   87.53   |   80.68    |   84.11   |   90.20   |   84.76    |   87.48   |   84.40   |   67.42   |   75.91   |
> |   W/o adaptation    |   71.20   |   13.27    |   42.24   |   43.75   |      6.27      |   25.01   |   9.13    |   22.31    |   15.72   |   49.25   |    4.13    |   26.69   |   14.72   |   5.33    |   10.03   |
> |     StarGAN v2      |   51.87   |   25.06    |   38.47   |   54.18   |     21.94      |   38.06   |   42.10   |    9.08    |   25.59   |   56.17   |   15.14    |   35.66   |   59.38   |   20.32   |   39.85   |
> |       PSTUDA        | **83.88** | **73.88**  | **78.88** | **82.89** |   **77.08**    | **79.99** | **75.84** | **59.73**  | **67.79** | **84.37** | **72.15**  | **78.26** | **82.28** | **68.58** | **75.43** |
>
> **W3 and Q2. Abdominal multi-organ segmentation.**
>
> Due to the limited availability of medical data, we apologize that we were unable to collect a multi-target domain abdominal multi-organ dataset within a short timeframe. However, we collected a multi-organ dataset [1-2] consisting of CT and MR images from internet to validate the proposed method. The experimental results are presented in the table below.
> |     **Methods**     |           |           | **CT→MR** |           |           |           |           | **MR→CT** |           |           |
> | :-----------------: | :-------: | :-------: | :-------: | :-------: | :-------: | :-------: | :-------: | :-------: | :-------: | :-------: |
> |                     |   Liver   | R. kidney | L. kidney |  Spleen   |   Avg.    |   Liver   | R. kidney | L. kidney |  Spleen   |   Avg.    |
> | Supervised training |   94.39   |   90.86   |   73.38   |   78.00   |   84.16   |   87.45   |   69.33   |   77.76   |   75.61   |   77.54   |
> |   W/o adaptation    |   23.44   |   1.99    |   12.77   |   20.28   |   14.62   |   37.33   |   0.00    |   0.26    |   1.29    |   9.72    |
> |     StarGAN v2      |   75.04   |   68.16   |   62.83   |   68.13   |   68.54   |   44.32   |   28.05   |   26.51   |   24.85   |   30.93   |
> |       PSTUDA        | **88.15** | **74.20** | **70.51** | **71.99** | **76.21** | **53.54** | **53.42** | **60.28** | **38.48** | **51.43** |
>
> We observe that although the results of MR to CT showed an improvement of approximately 42% compared to no domain adaptation, there is still significant potential for enhancement in comparison to fully supervised methods. Furthermore, the results from CT to MR were better. We attribute this difference to the higher image quality of CT in contrast to MR, which presents a greater challenge for domain adaptation from MR to CT.
>
> **Q1 Insights and future work.**
>
> As shown in Table 8 of Appendix E, the ablation study indicates that PIN brings significant performance improvements. However, the results for the multi-scale discriminator in Table 7 show that the improvement is not as pronounced. In our future work, we will conduct further investigation into different versions of discriminators. Furthermore, PSTUDA is essentially a multi-domain to multi-domain translation framework. We will also focus on developing a dedicated one-to-multiple UDA framework to achieve more substantial performance improvements.
>
> **Q3 Feasibility of deployment.**
>
> By utilizing the generator and multi-level style dictionary from PSTUDA in a clinical environment, we can achieve rapid one-to-multiple domain adaptation. We conducted simulations of PSTUDA’s performance using minimal computational resources with clinical data from our partner hospitals. Our PSTUDA requires approximately 13M of storage and 15G of computational resources. Thus, we believe that PSTUDA can be easily deployed even in clinical settings with limited computational infrastructure.
>
> **Q4 Code open source.**
>
> We will release the code, pre-trained models, and detailed implementation guidelines upon acceptance.
>
> [1] Kavur, A. E., et al. CHAOS challenge-combined (CT-MR) healthy abdominal organ segmentation. MedIA 2021.
>
> [2] Landman, B., et al. Multi-Atlas Labeling Beyond the Cranial Vault. 2017.

---

> > ### Comment · Reviewer_pquR · 2024-08-07
> >
> > I appreciate the clarification on the complexities of the one-to-multiple UDA task. The trade-offs in coordinating complex mappings among multiple target domains and the resultant benefits in training time and resource costs are well-explained.
> >
> > The additional validation using the new T1 MRI sequences is adequate.
> >
> > Consider moving the ablation studies into the main paper.
> >
> > The simulations indicating the feasibility of deploying PSTUDA with minimal computational resources are reassuring. The specific resource requirements you provided are helpful for understanding the deployment context.
> >
> > I am pleased to hear about the plans to release the code, pre-trained models, and implementation guidelines upon acceptance.
> >
> > Overall, your rebuttal has adequately met my concerns. I maintain my score as it is.

---

> ### Author Response · Authors · 2024-08-09
> **Thanks for the feedback!**
>
> Thank you for your positive feedback on our manuscript. We're glad our efforts to address your questions were satisfactory. We will carefully consider your suggestion to move the ablation studies into the main paper.

---

### Official Review · Reviewer_tgyp · 2024-07-09

**Soundness:** 3
**Presentation:** 3
**Contribution:** 2
**Rating:** 4
**Confidence:** 4

**Summary:**

The authors propose a novel and efficient One-to-Multiple Progressive Style Transfer Unsupervised Domain-Adaptive (PSTUDA) framework to address the UDA task for MRI sequences. Specifically, they developed a multi-level style dictionary that explicitly stores style information for each target domain at different stages, reducing the burden on a single generator in multi-target transfer tasks and effectively decoupling content and style. Additionally, multiple cascaded style fusion modules are employed, utilizing point-wise instance normalization to progressively recombine content and style features, thereby enhancing cross-modal alignment and structural consistency. Experiments conducted on both the private MSKT and public KiTS19 datasets demonstrate the effectiveness of this approach.

**Strengths:**

1.	A multi-level style dictionary was developed, which reduces the burden on a single generator in multi-target transfer tasks and effectively decouples content and style.
2.	Multiple cascaded style fusion modules are employed to progressively recombine content and style features, thereby enhancing cross-modal alignment and structural consistency.
3.	The organization of the paper's structure is excellent.

**Weaknesses:**

1.	The open-source code is not explicitly provided.
2.	The explanation of the One-to-Multiple Framework is somewhat rough and needs more details.
3.	PSTUDA needs to be validated bidirectionally on cross-modal data, such as MRI and CT.

**Questions:**

1.	Reading content from the multi-level style dictionary, does it incur additional overhead? If a new unknown domain is encountered, the learning of the multi-level style dictionary requires a warm-up (there is additional overhead the first time a new domain is encountered).
2.	What is puzzling is that Table 6 shows the multi-level style dictionary only takes up 0.1M; does this include tensors?
3.	Is there any difference between the Generator and Discriminator Architecture in Section 3.4 and previous work?
4.	The theory of achieving independent style transfer for each pixel using Point-wise Instance Normalization (PIN) requires further validation. While it may be effective, the explanation of its application in style transfer is not sufficiently thorough. Additionally, although the targets in medical image segmentation are more precise, this does not necessarily imply that style adjustments need to be more precise. Further explanation is needed.

**Limitations:**

The authors have addressed the limitations of the current work.

---

> ### Author Rebuttal · Authors · 2024-08-07
>
> We greatly appreciate your constructive and insightful comments. We address all weaknesses and questions below.
>
> **W1. Open-source code is not explicitly provided.**
>
> We will release the code, pre-trained models, and detailed implementation guidelines upon acceptance. To ensure reproducibility, we build on the open-source implementations of StarGAN v2 and CycleGAN, and provide all relevant hyperparameters in Appendix B.
>
> **W2. One-to-Multiple framework needs more details.**
>
> We apologize for this and will provide a more detailed description of our One-to-Multiple PSTUDA framework in subsequent versions.
>
> **W3. Bidirectional validation on cross-modal MRI and CT data.**
>
> According to your suggestion, we conducted bidirectional cross-modal validation on the abdominal multi-organ dataset [1-2] and performed reverse validation experiments from MR to CT on the MSKT and KiT19 datasets. The results are shown in the following two tables. Both sets of experiments demonstrate that our method significantly outperforms StarGAN v2 (baseline).
>
> |     **Methods**     |           |           | **CT→MR** |           |           |           |           | **MR→CT** |           |           |
> | :-----------------: | :-------: | :-------: | :-------: | :-------: | :-------: | :-------: | :-------: | :-------: | :-------: | :-------: |
> |                     |   Liver   | R. kidney | L. kidney |  Spleen   |   Avg.    |   Liver   | R. kidney | L. kidney |  Spleen   |   Avg.    |
> | Supervised training |   94.39   |   90.86   |   73.38   |   78.00   |   84.16   |   87.45   |   69.33   |   77.76   |   75.61   |   77.54   |
> |   W/o adaptation    |   23.44   |   1.99    |   12.77   |   20.28   |   14.62   |   37.33   |   0.00    |   0.26    |   1.29    |   9.72    |
> |     StarGAN v2      |   75.04   |   68.16   |   62.83   |   68.13   |   68.54   |   44.32   |   28.05   |   26.51   |   24.85   |   30.93   |
> |       PSTUDA        | **88.15** | **74.20** | **70.51** | **71.99** | **76.21** | **53.54** | **53.42** | **60.28** | **38.48** | **51.43** |
>
> |     **Methods**     |           | **MR(T1c)→CT** |           |           | **MR(FS T2W)→CT** |           |
> | :-----------------: | :-------: | :------------: | :-------: | :-------: | :---------------: | :-------: |
> |                     |  Kidney   |     Tumor      |   Avg.    |  Kidney   |       Tumor       |   Avg.    |
> | Supervised training |   91.98   |     66.61      |   79.30   |   91.98   |       66.61       |   79.30   |
> |   W/o adaptation    |   41.87   |     16.26      |   29.07   |   8.07    |       11.36       |   9.72    |
> |     StarGAN v2      |   34.44   |     24.02      |   29.23   |   46.17   |       22.96       |   34.57   |
> |       PSTUDA        | **73.13** |   **56.23**    | **64.68** | **78.13** |     **50.85**     | **64.49** |
>
> **Q4. Theory of PIN.**
>
> We believe that PIN has significant advantages in certain complex scenarios. For example, in kidney tumor medical imaging data, the contrast, texture, and morphology of the lesion areas exhibit significant differences from normal tissues. These differences ultimately stem from variations in data distribution. In such cases, PIN customizes a set of scaling parameters for each local spatial point in the content feature map for each channel, allowing for more precise alignment of local differences, thereby making the translated images closer to the real target domain’s data distribution. The calculation formula for PIN is:
> $$
> \huge y_{nchw} = \gamma_{nchw}(v_{ss}) \cdot \hat x_{nchw} + \beta_{nchw}(v_{ss})
> $$
> $$
> \huge \hat x_{nchw} = \frac{x_{nchw} - \mu_{nc}}{\sqrt{\sigma_{nc}^2 + \epsilon}}
> $$
> $$
> \huge \gamma_{nchw}(v_{ss}), \beta_{nchw}(v_{ss}) = chunk(h_{n(2c)hw})
> $$
> $$
> \huge h_{n(2c)hw} = ConvBlock(v_{ss})
> $$
> where $\gamma$ and $\beta$ are obtained through convolutional transformations via $v_{ss}$, and their dimensions are consistent with those of $\hat x_{nchw}$.
>
> **Q4. Effectiveness and Interpretation of PIN.**
>
> We conducted a comparison of PIN with the widely used AdaIN [3] and BIN [4], as shown in the table below. The results show that our PIN achieves the best performance. Theoretically, fully supervised segmentation results should be the upper limit for UDA, as it is conducted within the same domain, with both the training and test sets belonging to the same distribution. The key challenge in UDA tasks lies in addressing the differences in data distribution between different domains. Hence, the closer the distribution of the translated images is to the real target domain’s data distribution, the better the segmentation performance. Thus, we believe that the precise style adjustments in PIN will further reduce data distribution differences, leading to more accurate target segmentation.
>
> | **Normalization** |  | **T1c→FS T2W** |  |  | **T1c→T2W** |  |  | **T1c→DWI** |  |
> |:---:|:---:|:---:|:---:|:---:|:---:|:---:|:---:|:---:|:---:|
> |  | Kidney | Tumor | Avg. | Kidney | Tumor | Avg. | Kidney | Tumor | Avg.  |
> | AdaIN | 85.05  | 62.11  | 73.58  | 75.40  | 43.32  | 59.36  | 83.69  | 64.44  | 74.07   |
> | BIN | 82.32  | 67.91  | 75.12  | 74.14  | 49.02  | 61.58  | 85.85  | 65.07  | 75.46   |
> | PIN | **86.30** | **76.36** | **81.33** | **77.26** | **53.77** | **65.52** | **86.99** | **74.23** | **80.61** |
>
> **Additional comment:** We would have liked to include the rest of answers to the questions mentioned by Reviewer **tgyp** (marked as **Q1**,**Q2**, and **Q3**). Unfortunately, we did not have enough space in this rebuttal box. As soon as the discussion phase will begin, we will include the mentioned answers in an additional comment for the reviewer.

---

> ### Author Response · Authors · 2024-08-07
> **Response to additional questions**
>
> As we mentioned in the main rebuttal, we include the answers to the additional questions of Reviewer **tgyp**. We hope that this helps to address all remaining concerns and we thank again for taking the time to review our work.
>
> **Q1. Overhead and warm-up for multi-level style dictionary.**
>
> In our experiments, the multi-level style dictionary essentially consists of a set of learnable tensors. Content is accessed through indexing, and as the style vectors are independent of each other, this does not result in any additional overhead.
>
> Our method primarily focuses on adapting a source domain to multiple target domains simultaneously. All domains are fixed and visible during training, so there are no encounters with new unknown domains. Additionally, the multi-level style dictionary is randomly initialized at the beginning of training and does not require a warm-up phase.
>
> **Q2. Size of Multi-level Style Dictionary.**
>
> In our experiments, the multi-level style dictionary is directly created as a set of learnable tensors. Specifically, its dimensions are (4, 6, 4096), where 4 represents the number of domains, 6 represents the number of style fusion layers (four layers in the style fusion module and two layers in the decoder), and 4,096 represents the depth of the style dictionary. Therefore, this tensor contains a total of 4×6×4096=98,304 parameters, which converts to approximately 0.10M.
>
> **Q3. Generator and Discriminator Architecture.**
>
> * **Generator:** PSTUDA uses the CycleGAN generator. The difference lies in the replacement of normalization operations in the style fusion module and decoder from AdaIN to our proposed PIN. Additionally, we employ a progressive style injection method layer by layer to achieve image translation.
>
> * **Discriminator**: PSTUDA’s discriminator architecture is based on the multi-scale discriminator from MUNIT. The difference is that we replace the Conv2dBlock in MUNIT with ResBlk, which has a residual structure. Additionally, the multi-scale output sizes are set to 1/16, 1/32, 1/64, and 1/128 of the original image size. Our motivation for using a multi-scale discriminator is that the generator, integrated with PIN, will have enhanced generative performance. Therefore, the discriminator also needs to be more powerful to match the generator, enabling better adversarial training and effectively leveraging the generator’s capabilities.
>
> [1] Kavur, A. E., et al. CHAOS challenge-combined (CT-MR) healthy abdominal organ segmentation. MedIA 2021.
>
> [2] Landman, B., et al. Multi-Atlas Labeling Beyond the Cranial Vault. 2017.
>
> [3] Huang, X., et al. Arbitrary Style Transfer in Real-time with Adaptive Instance Normalization. ICCV 2017.
>
> [4] Nam, H., et al. Batch-Instance Normalization for Adaptively Style-Invariant Neural Networks. NeurIPS 2018

---

> > ### Comment · Reviewer_tgyp · 2024-08-12
> >
> > The authors provided bidirectional cross-modal experimental results, further demonstrating the generalization capability of the method. Additionally, they addressed each of the raised questions individually and promised to make revisions in the revised version. Therefore, I suggest changing the score to 5: Borderline Accept.
> >
> > Furthermore, there is a minor issue: I hope the authors can specify the precision used in the model, such as whether it is FP16, in the revised version.

---

> > > ### Author Response · Authors · 2024-08-13
> > > **Thanks for the feedback!**
> > >
> > > Thank you for recognizing our efforts in the rebuttal. We greatly appreciate your consideration in raising the rating of our paper. Your feedback has been invaluable to our work. Based on your suggestions, we will specifically clarify the precision used in the model in the revised version.

---

### Official Review · Reviewer_yJwR · 2024-07-14

**Soundness:** 2
**Presentation:** 3
**Contribution:** 3
**Rating:** 5
**Confidence:** 4

**Summary:**

The paper presents a one-to-multiple progressive style transfer unsupervised domain adaptation framework designed for kidney and tumor segmentation.

It aims to mitigate the challenges of annotation burden and domain differences by employing a multi-level style dictionary and cascading style fusion modules.

It demonstrates significant improvements in segmentation performance and efficiency on the MSKT and KiTS19 datasets.

**Strengths:**

x. The introduction of a multi-level style dictionary and point-wise instance normalization (PIN) for progressive style transfer that effectively decouples content and style features.

x. It significantly reduces the floating-point computation by approximately 72% and the number of model parameters by about 50%, highlighting its efficiency and feasibility for practical clinical applications.

**Weaknesses:**

x. **Concern about novelty.**

First, the authors extended the UDA setting to 'one-to-multiple'. I am not sure about its practical or clinical relevance here compared to continued UDA or UDA on evolving domains.
Second, there seems to be limited novelty compared to the architecture of OMUDA derived from StarGAN v2.

x. **Concern about the effectiveness of PIN**:
Point-wise Instance Normalization is a central component of the framework, but its effectiveness compared to other normalization techniques is not thoroughly evaluated. There might be scenarios where the PIN is less effective, and alternative normalization methods like Adaptive Instance Normalization or Batch-Instance Normalization might perform better.

x. **Concern on analysis.**
The multi-level style dictionary and the cascading style fusion modules. Understanding the contribution of each part would help in refining the model. The sensitivity of the model’s performance to various hyperparameters, such as the temperature parameter (\tao), the number of style fusion modules, and the depth of the style dictionary, is not extensively studied.

**Questions:**

Please see my comments above.

**Limitations:**

yes

---

> ### Author Rebuttal · Authors · 2024-08-07
>
> We greatly appreciate your constructive and insightful comments. We address all weaknesses below.
>
> **W1. Practical and clinical relevance of PSTUDA.**
>
> PSTUDA is designed as a one-to-multiple UDA framework based on multi-sequence MRI segmentation tasks, thus having strong clinical relevance. Compared to methods that train target domains sequentially (including continued UDA and UDA on evolving domains), PSTUDA offers the following advantages:
>
> * **Flexibility:** Sequential methods treat all target domains as a single domain during training, which prevents them from generating images of the specified target domain during inference, potentially leading to inaccurate results. In contrast, PSTUDA can generate images of the specified target domain as needed during inference. This flexibility meets the diverse requirements of clinical applications.
> * **Training efficiency:** Sequential methods train in a serial manner, whereas PSTUDA trains all source and target domains in parallel, significantly reducing training time. Additionally, PSTUDA’s generator and multi-level style dictionary enable quick deployment and fast inference, meeting clinical demands.
>
> **W1. Novelty of PSTUDA.**
>
> PSTUDA is inspired by OMUDA and StarGAN v2. However, we believe there are some fundamental differences between our method and these models:
>
> * **Style feature acquisition:** OMUDA extracts style features from a style encoder or mapping network, which lacks representativeness and stability. In contrast, PSTUDA addresses this issue by using a multi-level style dictionary to directly learn the global style features of each domain.
> * **Generator architecture:** Both OMUDA and PSTUDA use the CycleGAN generator. However, OMUDA uses the same style features in each layer of AdaIN, while PSTUDA replaces AdaIN with PIN and progressively injects different style features layer by layer to achieve image translation.
> * **Discriminator architecture:** OMUDA uses the StarGAN v2 discriminator, whereas PSTUDA employs a multi-scale discriminator to better leverage the capabilities of its powerful generator integrated with PIN.
>
> The ablation experiments in Appendix E demonstrate the effectiveness of our method.
>
> **W2. Evaluation of PIN effectiveness.**
>
>  AdaIN [1] and BIN [2] provide global scaling parameters for each channel of the content feature map. In contrast, our PIN offers unique scaling parameters (mean and standard deviation) for each local spatial point in each channel.
>
> We believe that PIN is particularly valuable for fine-grained segmentation tasks due to its ability to consider local style differences, enabling PSTUDA to generate synthetic images that better match the target domain data distribution.
>
> To demonstrate our perspective, we conducted ablation experiments on the three methods, as presented in the table below. The results indicate that PIN outperforms AdaIN and BIN in terms of performance, especially in kidney tumor segmentation. This superiority can be attributed to the fact that kidney tumors, being abnormal pathological tissues, exhibit significant style differences which can be finely processed by PIN.
>
> | **Normalization** |           | **T1c→FS T2W** |           |           | **T1c→T2W** |           |           | **T1c→DWI** |           |
> | :---------------: | :-------: | :------------: | :-------: | :-------: | :---------: | :-------: | :-------: | :---------: | :-------: |
> |                   |  Kidney   |     Tumor      |   Avg.    |  Kidney   |    Tumor    |   Avg.    |  Kidney   |    Tumor    |   Avg.    |
> |       AdaIN       |   85.05   |     62.11      |   73.58   |   75.40   |    43.32    |   59.36   |   83.69   |    64.44    |   74.07   |
> |        BIN        |   82.32   |     67.91      |   75.12   |   74.14   |    49.02    |   61.58   |   85.85   |    65.07    |   75.46   |
> |        PIN        | **86.30** |   **76.36**    | **81.33** | **77.26** |  **53.77**  | **65.52** | **86.99** |  **74.23**  | **80.61** |
>
> **Additional comment:** We would have liked to include the rest of answers to the questions mentioned by Reviewer **yJwR** (marked as **W3**). Unfortunately, we did not have enough space in this rebuttal box. As soon as the discussion phase will begin, we will include the mentioned answers in an additional comment for the reviewer.

---

> > ### Comment · Reviewer_yJwR · 2024-08-13
> > **thank you.**
> >
> > Most of my concerns are addressed. Hence I raised the score to 5.
> > Please include the new results in the manuscript if it is accepted.

---

> ### Author Response · Authors · 2024-08-07
> **Response to additional questions**
>
> As we mentioned in the main rebuttal, we include the answers to the additional questions of Reviewer **yJwR**. We hope that this helps to address all remaining concerns and we thank again for taking the time to review our work.
>
> **W3. Ablation Study on key hyperparameters affecting model performance.**
>
>  Our main contributions lie in the multi-level style dictionary and the style fusion module. These two modules have the following hyperparameters: the depth of the style dictionary, the number of levels in the style dictionary, and the number of style fusion modules. Note that the number of levels in the style dictionary is equal to the number of style fusion modules.
>
> To investigate the sensitivity of model performance to various hyperparameters, according to your suggestions, we conducted extensive ablation studies on the depth of the multi-level style dictionary (MSD) and the number of style fusion modules (SFM). The results, presented in the table below, indicate that under the original settings (dictionary depth of 4,096, module number of 4), PSTUDA performs optimally in most experiments. Notably, when we extended our dictionary depth to 16,384, there was an improvement in segmentation performance on T2W sequence.
>
> | **MSD Depth** |           | **T1c→FS T2W** |           |           | **T1c→T2W** |           |           | **T1c→DWI** |           |
> | :-----------: | :-------: | :------------: | :-------: | :-------: | :---------: | :-------: | :-------: | :---------: | :-------: |
> |               |  Kidney   |     Tumor      |   Avg.    |  Kidney   |    Tumor    |   Avg.    |  Kidney   |    Tumor    |   Avg.    |
> |      256      |   84.85   |     66.34      |   75.60   |   70.55   |    43.85    |   57.20   |   83.15   |    63.52    |   73.34   |
> |     1024      |   85.84   |     69.68      |   77.76   |   70.06   |    42.76    |   56.41   |   83.99   |    64.41    |   74.20   |
> |     4096      | **86.30** |   **76.36**    | **81.33** |   77.26   |    53.77    |   65.52   | **86.99** |  **74.23**  | **80.61** |
> |     16384     |   84.63   |     66.74      |   75.69   | **77.63** |  **57.49**  | **67.56** |   83.31   |    68.78    |   76.05   |
>
> | **SFM Number** |           | **T1c→FS T2W** |           |           | **T1c→T2W** |           |           | **T1c→DWI** |           |
> | :------------: | :-------: | :------------: | :-------: | :-------: | :---------: | :-------: | :-------: | :---------: | :-------: |
> |                |  Kidney   |     Tumor      |   Avg.    |  Kidney   |    Tumor    |   Avg.    |  Kidney   |    Tumor    |   Avg.    |
> |       2        |   85.65   |     71.77      |   78.71   |   73.94   |    44.51    |   59.23   |   83.89   |    61.31    |   72.60   |
> |       4        | **86.30** |   **76.36**    | **81.33** | **77.26** |  **53.77**  | **65.52** | **86.99** |  **74.23**  | **80.61** |
> |       6        |   84.49   |     67.94      |   76.22   |   74.97   |    50.57    |   62.77   |   81.20   |    60.44    |   70.82   |
> |       8        |   84.09   |     63.86      |   73.98   |   60.06   |    30.43    |   45.25   |   83.75   |    60.60    |   72.18   |
>
> [1] Huang, X., et al. Arbitrary Style Transfer in Real-time with Adaptive Instance Normalization. ICCV 2017.
>
> [2] Nam, H., et al. Batch-Instance Normalization for Adaptively Style-Invariant Neural Networks. NeurIPS 2018.

---

> ### Author Response · Authors · 2024-08-14
> **Thanks for the feedback!**
>
> We are pleased to have addressed most of your concerns and we sincerely appreciate you raising our paper's score to 5: Borderline Accept. Your feedback has been invaluable to our work, and based on your suggestions, we will include the new supplementary experimental results in the revised version. Thank you for your support！

---

### Author Rebuttal · Authors · 2024-08-07

Dear reviewers and AC,

We sincerely appreciate your valuable time and effort spent reviewing our manuscript. We would like to thank all the reviewers for providing insightful comments and valuable suggestions. The valuable feedback from the reviewers has significantly contributed to enhancing the quality of our manuscript.

Based on the comments from the reviewers, we have summarized the strengths of our paper as follows:

* **Method: [Reviewer yJwR, tgyp, pquR]** PSTUDA introduces a relatively interesting approach by employing cascaded style fusion modules and Point-wise Instance Normalization (PIN) to achieve excellent cross-modal alignment and structural consistency. The introduction of a multi-level style dictionary and point-wise instance normalization (PIN) for progressive style transfer that effectively decouples content and style features. A multi-level style dictionary was developed, which reduces the burden on a single generator in multi-target transfer tasks and effectively decouples content and style. Multiple cascaded style fusion modules are employed to progressively recombine content and style features, thereby enhancing cross-modal alignment and structural consistency.

* **Experiment: [Reviewer yJwR, pquR]** It significantly reduces the floating-point computation by approximately 72% and the number of model parameters by about 50%, highlighting its efficiency and feasibility for practical clinical applications. PSTUDA’s experimental design is robust, featuring both comparative and ablation studies as well as appropriate metrics and analysis. This paper is original, detailed, clear, and demonstrates reasonable improvements in segmentation with significant improvements in floating point operations and model size.

* **Expression: [Reviewer tgyp, pquR]** The organization of the paper's structure is excellent. The paper is well-written and clear, providing detailed explanations of the architecture and methodologies. The comprehensive writing extends to the appendices, offering an appropriate level of detail on datasets, which aids in reproducibility and understanding.

* **Impact: [Reviewer pquR]** It emphasizes structural consistency, which is critical for medical image analysis. The strong review of related works situates PSTUDA within the broader context of domain adaptation and medical image segmentation. This combo allows for more precise and progressive recombination of content and style features, addressing limitations in existing one-to-one UDA methods and extending the applicability of domain adaptation techniques in medical imaging.

We summarized our novelty as follows:

* Unsupervised domain adaptation is a mainstream method for addressing domain distribution discrepancies. However, existing UDA methods are mostly limited to one-to-one domain adaptation, which is often inefficient and resource-intensive when dealing with multi-sequence medical image domain adaptation tasks. To address this challenge, we propose a novel and efficient one-to-multiple progressive style transfer UDA framework. It uses a single generator to simultaneously translate one source domain to multiple specified target domains. Compared to one-to-one methods, our approach not only achieves optimal performance but also significantly reduces model parameters and floating-point computation, highlighting its efficiency and feasibility in practical clinical applications.

* We developed a multi-level style dictionary to explicitly store the style information of each target domain at different stages, which alleviates the burden on a single generator in multi-target domain adaptation tasks and achieves effective decoupling of content and style.

* We employed multiple cascaded style fusion modules that progressively utilize point-wise instance normalization to inject local styles into content features. This fine-grained style transfer further reduces domain discrepancies and enhances cross-modal alignment and structural consistency.

Additionally, we have completed the theoretical proof and experimental validation of PIN, bidirectional cross-modal experiments for abdominal multi-organ segmentation, generalization experiments for more target domains, and all supplementary ablation experiments as requested by the reviewers.

We addressed each comment and question in detail below, and we kindly request the Reviewers **yJwR** and **tgyp** to reconsider our work in light of these aspects and support our efforts. If there are any further questions or concerns, we would be happy to discuss them with you.

We strongly believe that PSTUDA can be a useful addition to the NeurIPS community and above innovative contributions elevate the value and significance of our research in the realm of multi-target medical image domain adaptation and provide useful references and insights for future research.

Thank you very much!



Best regards,

Authors.

---

### Decision · Program_Chairs · 2024-09-25

**Decision:**

Accept (poster)

**Comment:**

The paper receives positive reviewing comments from three reviewers. Two reviewers rate it as 'borderline accept' and a third reviewer rates it as 'strong accept' after rebuttal and discussions.

The rating by Reviewer tgyp is shown as 'Borderline reject', but in the discussion box, s/he mentioned explicitly that "I suggest changing the score to 5: Borderline Accept."

Therefore, I recommend 'Accept'.